# Adaptation to Spanish and psychometric study of the Flow State Scale-2 in the field of musical performers

**Laura Moral-Bofill**(ID)**, Andrés Lópezdelallave, Mª. Carmen Pérez-Llantada, Francisco Pablo Holgado-Tello**(ID)**\***

Department of Methodology of the Behavioral Sciences, Faculty of Psychology, Universidad Nacional de Educación a Distancia (UNED), Madrid, Spain

\* pfholgado@psi.uned.es

## Abstract

Flow is a positive and optimal state of mind, during which people are highly motivated and absorbed in the activity they are doing. It is an experience that can occur in any area of life. One of the measurement instruments which is most commonly used to evaluate this construct is the Flow State Scale-2 (FSS-2). This instrument has been used in different languages, mainly in the field of sport. In this research work, the FSS-2 has been translated into Spanish and administered to 486 musicians from different regions of Spain in order to examine the psychometric properties. A version which uses six dimensions from the original questionnaire has been used—those that constitute the experience of flow—and four alternative models have been analysed (Six related factors model, two second order factor models and a bifactor model). The results revealed that the dimension *time* could be controversial and more research could be needed. In general terms, the six-factor model (RMSEA = .04; GFI = .99; AGFI = .99) and a second-factor one (RMSEA = .033; GFI = .99; AGFI = .99) are solutions consistent with previous studies and show that the questionnaire can be considered a reliable (Alphas of the dimensions range from .81 to .94; Omegas from .86 to .97; and mean discrimination of the dimensions from .64 to .88) and useful tool, both in clinical and educational contexts, as well as an instrument for the evaluation of this construct in future research. However, the results of this study also suggest that flow can be explored in greater depth in musicians, taking into account that the writing of the original items was based on the experience of athletes and that the role of *time* in flow needs to be investigated.

## Introduction

The concept of flow has aroused much interest in recent years, leading to a great deal of research related to this topic. According to Csikszentmihalyi [1], flow is a positive and optimal state of mind during which people are highly motivated and absorbed in the activity they are doing, and it is an experience that can occur in any area of life. For the conceptualisation of

**Funding:** The authors received no specific funding for this work.

**Competing interests:** The authors have declared that no competing interests exist.

flow, Csikszentmihalyi [1] used a phenomenological model of consciousness based on information theory. Based on this approach, it was proposed that a mental event could be better understood when the way it was being experienced was directly observed. The optimal experience, or flow, would occur when the information that reaches consciousness is consistent with the goals set. In this way, the activity flows effortlessly and there is no reason to question one's own ability, while there is at the same time a positive feedback and a strengthening of self-confidence. The experience of flow is significant because it is a determining factor in order to make the present moment more enjoyable, but also because it strengthens self-confidence and advances the development of personal skills.

Csikszentmihalyi proposed a multidimensional model which, based on what he calls "the phenomenology of enjoyment", defines the state of flow by means of the following eight components [1]:

- Challenge-skill balance (1). This dimension aims to apprehend the balance between one's own abilities and the objective to be achieved.

- Concentration on the task (2). This refers to the way the capacity for attentional self-control is manifested while the activity being evaluated is in process.

- Clear goals and clear feedback (3 and 4). The goals which one is looking to achieve are unequivocally established and, during the completion of the task, their gradual achievement can be evaluated in a reliable and precise manner.

- Action-awareness fusion (5). This indicates the synchrony between awareness and action. From a phenomenological perspective, it implies the perception that one is acting without effort, with a deep involvement that moves the concerns and pressures of daily life away from consciousness.

- Sense of control (6). This refers to the presence of a feeling of control over one's actions, or more accurately, it implies a lack of concern regarding losing control.

- Loss of self-consciousness (7). This indicates the degree to which the task is absorbing and central at that existential moment. In relation to this experience there is a lack of concern for any other interest or concern which, at other times, could be central to the person's life.

- Transformation of time (8). This dimension refers to the altered perception of the passing of time while the activity is taking place.

The combination of all these elements leads to a deep feeling of enjoyment that rewards the person, what he calls: autotelic experience [1]. This is a distinctive and fundamental factor of the flow experience and describes the highly positive emotional value of this experience.

Based on this theoretical model, research into the flow experience has been conducted following different procedures and using different measurement techniques. One of the lines that has generated a large amount of research is that which has been carried out in the field of sport. In this context, a scale has been proposed in order to measure the optimal experience, the Flow State Scale (FSS) [2, 3], which has been used in numerous research projects. The FSS is an instrument that enables one to evaluate the eight theoretical dimensions outlined above, but which adds a scale in order to measure the autotelic experience. It therefore presents a questionnaire which proposes nine factors. In addition, the scale, taken as a whole, can be considered as a tool for measuring "global flow".

Subsequently, in order to improve the psychometric properties of the scale, the FSS-2 questionnaire was developed [4, 5]. The FSS-2 questionnaire was designed in order to evaluate the flow experience in the context of physical activity, but it has also been used for the evaluation

of flow in other activities such as musical performance [5]. The scales that compose it provide values for internal consistency of between .83 and .92. The goodness-of-fit indices are considered good for two proposed models: a model of nine first-order factors (Comparative Fit Index, CFI = .939; Chi-Square Non-Normed Fit Index, NNFI = .931; Root Mean Square Error of Approximation, RMSEA = .051) and a hierarchical model with nine first-order factors and one second-order factor (CFI = .93; NNFI = .920; RMSEA = .054). Only the "transformation of time" dimension presents a factor loading which is relatively low (.23) to one second-order factor ("global flow"). The authors [5] thus suggest that the "transformation of time" factor not be taken into account when calculating global flow.

Subsequently, a reduced version of FSS-2 [6] was developed: the Short State Flow Scale, consisting of nine elements. This version of the FSS-2 has been validated using a varied sample of activities, including musical performance.

Currently, research can be found that addresses adaptations and validations, in different languages and countries, both, of the original scale, the FSS, and the improved one, the FSS-2 [7, 8, 9, 10, 11, 12]. These studies regarding the adaptation of the FSS have also been carried out in Spanish [13]. However, these adaptations and validations have been carried out mainly in the field of sport, ignoring other situations in which the experience of flow could also be considered of interest.

Given the potential utility of the flow construct, in order to understand the state of optimal experience, the need to research it in musical activity, particularly in musical performers, has also been considered. Thus, Wrigley and Emmerson examined the properties of FSS-2 in a sample of Australian musicians [14]. Their results showed internal consistency values of between .81 and .92 for the scales of the instrument. The goodness-of-fit indices showed good results for the model of nine first-order factors (CFI = .96; Tucker-Lewis Index, TLI = .96; RMSEA = .04). In the case of the hierarchical model with nine first-order factors and one second-order factor, all the scales predicted the flow state, with all the regression weights within a significant critical proportion. In almost all cases the beta values exceeded .30, varying between .46 and .85. Only one exception was found to these results, and that is in relation to the "transformation of time" factor, a dimension which, as has been mentioned previously when presenting research in the field of sport, repeatedly shows low values.

Flow has been linked to increases in motivation, improved competition, and growth of individual abilities [15]. It has been pointed out that it contributes to improvement in terms of the technical and expressive training of musicians [16], while at the same time determining an increase in the time devoted to musical practice [17]. Therefore, although the FSS-2 is not a specific measuring instrument used to evaluate the experience of flow in musicians, numerous studies have used it in the context of musical performance. Thus, the flow experience has been studied as a desirable state of musicians during their performances, because it could enhance the quality and positive experience of the performance [14]. Other research has used the FSS-2 to assess the experience of the flow state in musicians and their conclusions contain certain implications for the development of musical learning [18]. Flow state has also been related to performance anxiety [18, 19], emotional intelligence [20], to the style of the music being performed [21], and the situation in which the musical performance takes place [21]. Certain aspects of the environment which may be facilitators or inhibitors of the flow experience have also been described [22]. Finally, some researchers have focused on flow from a social perspective, which considers not only optimal performance, but also the optimal interaction between two or more people [23, 24].

Recently, flow theory has begun to distinguish between the conditions that are necessary to give rise to flow and the psychological components that constitute the experience of flow [25, 26]. Therefore, in order to enter flow, an appropriate balance between the skills and the

challenges that a person faces is deemed necessary, as well as having clear objectives which are proximate to the action and, also, that there is clear and immediate feedback. The other dimensions would be the subjective experience of the flow state: concentration on the task, action-awareness, sense of control, loss of self-consciousness, transformation of time and autotelic experience [25, 26]. This research is conducted within this theoretical framework, in line with other authors who have measured flow state in musicians taking this reformulation into account [18].

In view of all of the above, the need to adapt the FSS-2 to Spanish and to look at the psychometric properties of the instrument in a population composed specifically of musicians was considered. The aim was to have a tool available which could be used to assess flow state in musicians. On the one hand, it was hypothesised that the results of the psychometric analysis of the FSS-2 scale adapted and translated into Spanish in a population of performing musicians would be consistent with the results obtained in other studies that have examined the psychometric properties of the FSS-2. On the other hand, a second hypothesis was presented according to which, if the factors of skill-challenge balance, clear objectives and clear feedback are conditions in order to enter the flow state, they will positively correlate with the other factors that represent the flow experience or state. This correlation can therefore be deemed as supporting the criterion validity.

The results of this study share many similarities with those of other research projects [8, 4, 9, 10, 14]. The six scales considered as well as the global flow scale present values that indicate good internal consistency and discrimination. Moreover, the structural models analysed present good goodness-of-fit indices for a model of six first-order related factors and a hierarchical model of six first-order factors and one second-order. Following the theoretical model indicated, we have used the three FSS-2 scales that measure the preconditions to enter flow as criteria measurements and to see how they relate to the six scales that measure flow state. The results show how the scale that measures the goals for the action, once partial correlations are made and the effect of the other two conditions for flow is controlled, goes from being significantly correlated with all the dimensions of the questionnaire (except the dimension that measures transformation of time) to not be significant and practically zero.

## Materials and methods

The research has been carried out following the standards recommended for research on human participants from the code of ethics of the European Community and the American Psychological Association´s Ethical Standards for Research and Publication. The research was approved by the Bioethics Committee of the UNED. We have guaranteed privacy in the processing of data. Participation in the study was voluntary and anonymous.

### Participants

A sample of 558 participants was obtained, including music students as well as amateur and professional musicians. They were all Spanish speakers and came from different regions of Spain. As a criterion for inclusion, it was established that participants would have a well-established relationship with musical performance (students, professionals, amateurs), specifically at least two years of study; as a criterion for exclusion a minimum age of 18 was established. As a consequence of these criteria, 72 persons were excluded, so in the end the total number of participants was 486, with an age range between 18 and 83 years old (mean 38.17 and $SD = 12.91$). Men accounted for 38.90% of the sample (mean age 38.91; $SD = 12.97$), while 60.50% were women (mean age 37.77; $SD = 12.90$). Three participants preferred not to answer

this questionnaire (0.6%; mean age 31.33; *SD* = 9.07). Participation in the study was voluntary, with no financial or academic reward.

## Procedure

The sample was obtained by means of snowball sampling. Through the UNED's social networks and communications tools, all interested persons were offered the possibility of participating in this research by completing a survey that was submitted online. The form was published via the Google Forms tool, in which the EFIM questionnaire was included. The questions were organised in such a way that it was "mandatory" to answer all of them (Google signals this requirement with a red asterisk at the end of each question). The participants thus answered all the questions in the survey and there were no cases where the answers to any of the questions set by the tool were missing. The time required to complete the survey was approximately 15 minutes. Addressees were informed that participation, which was anonymous and voluntary, consisted of filling out a Google form, in which the Spanish adaptation of the FSS-2, the scale "Estado de Fluidez para Intérpretes Musicales" (EFIM) was included.

## Instruments

- *The scale "Estado de Fluidez para Intérpretes Musicales", EFIM* (https://www.mindgarden. com/100-flow-scales#horizontalTab2). This is a 24-item questionnaire that measures the Flow State. It consists of six scales, each with four items and conceptually different: *Action- awareness merging (it will be merging); Total concentration on the task at hand (concentration); Sense of control (control); Loss of self-consciousness* (*consciousness*); *Transformation of time (time); and Autotelic experience (autotelic)*. To assess the degree of agreement with the formulation of each element, on the original scale, the FSS-2, a Likert scale with five anchor points (1 to 5) [5] was used. However, reports have been submitted indicating that larger amplitudes of the scale appear to improve the sensitivity and accuracy of the measurements [27, 28, 29, 30, 31], while other reports challenge the use of the central categories in these types of scales, such as 3 in the case of using a scale of 1 to 5, suggesting that it may affect both the accuracy of the measurements and the validity of the inferences made [32, 33, 34, 35, 36]. In addition, in our cultural field it is usual and widespread to use scales from 0 to 10 when almost any object or event has to be evaluated or assessed [37]. Due to these considerations, the questionnaire that was presented to the participants was answered on a Likert scale of 0 to 10 points. On the other hand, when using these types of scales, it is suggested that verbal labels associated with extreme scores be included, so that they guide the trend of the values: zero is labelled with a "totally disagree" and ten with "totally agree" [37], so that, the higher the score, the greater the flow state. The Spanish version of this questionnaire was developed in accordance with the guidelines of the International Test Commission [38], and using the back-translation method based on the original English version: 1) the original version was translated into Spanish by a bilingual group expert in psychology; 2) the new version was translated back into English by a different translator, also bilingual and a psychologist; and 3) the discrepancies arising were discussed and the appropriate corrections made to the new version of the EFIM. With the form, which included socio-demographic questions in addition to the EFIM, a pilot test was carried out with 20 musicians. The results of this test were satisfactory since the respondents did not report any difficulties understanding the questionnaire.

- The remaining 3 scales of the FSS-2 that were not included in the EFIM were used as criteria, as they correspond to the dimensions that are necessary conditions in order to generate the flow state. These are: *Challenge-skill balance* (*balance*) (in this sample: Alpha = .75; Omega =

.83; mean discrimination = .59); *Clear goals (goals)* (Alpha = .90; Omega = .90; mean discrimination = .78); *Unambiguous feedback (feedback)* (Alpha = .86; Omega = .87; mean discrimination = .56).

## Statistical analyses

In order to obtain evidence of construct validity of the instrument in a Spanish sample, we tested the original model proposed by Jackson and Eklund [4] using the Confirmatory Factor Analysis (CFA) procedure [39]. After analysing the goodness-of-fit indices and the patterns of correlations between the latent variables, we also tested alternative models.

Although the items are ordinal in nature, given that there are 11 response options, the univariate normality test for asymmetry and kurtosis was analysed in order to guide the election of the estimation method which was most suitable [40, 41, 42]. We used polychoric correlations (see S1 File), and as an estimation method that of Robust Unweighted Least Squares (RULS), given the large number of variables and the distribution of the items [40, 43, 44].

The statistical analyses were carried out using the following applications: PRELIS 2.30, LISREL 8.8 [45, 46] and SPSS 24.0.0.0 [47].

## Results

### Descriptive statistics

Table 1 shows the basic descriptive statistics of the items. We would like to highlight that all the items have a negative skewness and, although there are 11 points in the response scale, neither of them has a normal distribution.

### Confirmatory factor analysis

According to the original structure proposed by Jackson and Eklund [4], the dimensions of the instrument are grouped into: *merging* (1, 7, 13, and 19); *concentration* (2, 8, 14 and 20); *control* (3, 9, 15, and 21); *consciousness* (4, 10, 16 and 22); *time* (5, 11, 17 and 23); and *autotelic* (6, 12, 18 and 24). It is thus a related six-factor model (Model 1).

As a result of carrying out a CFA on the model 1 (six related factors), the following global goodness-of-fit indices were obtained (see Table 2): $\chi^2$ (*d.f.* = 237; *p < .001*) = 341.46; RMSEA = .04 with an interval at 90% (.03 to .05) (values < 0.08 are adequate); GFI = .99; AGFI = .99; CFI = 1.00; TLI = 1.00; (values > 0.90 are adequate) SRMR = .04 (values < 0.10 are adequate); and BIC = -295.27 (the lower the better) [48].

The standardised solution for model 1 is shown in Table 3.

These results could be considered as a good fit. However, after inspecting the factor correlations (see Table 4), we found that *time* presents low relation with the other five factors of the models.

This result is consistent with previous studies that found that the relation between *time* and the rest of the dimensions is weak [8, 4, 9, 10, 14]. Supported by this result, it could be of interest to test a model with a second-order factor which explained these five factors, but don't explain *time* (model 2). The goodness-of-fit indices of this new model (Model 2) were: $\chi^2$ (*d.f.* = 238; *p < .001*) = 1292.39; RMSEA = .10 with an interval at 90% (.09 to .11); GFI = .92; AGFI = .90; CFI = .96; TLI = .95; SRMR = .14; BIC = 656.97. The significant increase of $\chi^2$ of 950.93 for 1 degree of freedom indicates that Model 2 is a significant deterioration on Model 1 (six related factors) (see Table 2). That is, Model 1 fits better than Model 2 (five first-order factors explained by one second-order factor except for time). Otherwise, the modification index suggests including the parameter that related *time* with the general second-order factor (flow).

**Table 1. Basic description of the items.**

| Items | Mean | SD | Skewness | Kurtosis | Normality test (Skewness and Kurtosis) Chi-Square | P-value |
|---|---|---|---|---|---|---|
| 1 | 7.00 | 2.49 | -.80 | .01 | 42.62 | < .01 |
| 2 | 7.94 | 1.90 | -.95 | .86 | 64.29 | < .01 |
| 3 | 7.27 | 2.05 | -.84 | .46 | 49.51 | < .01 |
| 4 | 6.49 | 2.93 | -.59 | -.63 | 43.49 | < .01 |
| 5 | 6.77 | 3.10 | -.89 | -.27 | 52.37 | < .01 |
| 6 | 8.25 | 1.94 | -1.66 | 3.23 | 162.01 | < .01 |
| 7 | 6.61 | 2.34 | -.59 | -.08 | 25.14 | < .01 |
| 8 | 7.27 | 2.45 | -.89 | .13 | 50.89 | < .01 |
| 9 | 7.37 | 2.09 | -.84 | .46 | 48.94 | < .01 |
| 10 | 6.20 | 2.94 | -.48 | -.84 | 65.48 | < .01 |
| 11 | 6.83 | 3.02 | -.93 | -.09 | 53.92 | < .01 |
| 12 | 7.87 | 2.34 | -1.34 | 1.40 | 108.01 | < .01 |
| 13 | 6.20 | 2.56 | -.50 | -.39 | 23.49 | < .01 |
| 14 | 7.79 | 1.94 | -.94 | .92 | 64.61 | < .01 |
| 15 | 6.73 | 2.39 | -.78 | .29 | 42.41 | < .01 |
| 16 | 6.12 | 2.94 | -.45 | -.84 | 62.47 | < .01 |
| 17 | 6.93 | 2.89 | -.93 | .04 | 54.08 | < .01 |
| 18 | 8.16 | 2.01 | -1.54 | 2.85 | 146.42 | < .01 |
| 19 | 6.52 | 2.40 | -.67 | .06 | 31.56 | < .01 |
| 20 | 7.72 | 1.99 | -1.03 | 1.10 | 74.96 | < .01 |
| 21 | 6.85 | 2.31 | -.72 | .08 | 35.85 | < .01 |
| 22 | 6.10 | 2.96 | -.39 | -.92 | 78.47 | < .01 |
| 23 | 6.78 | 2.89 | -.86 | -.09 | 47.71 | < .01 |
| 24 | 8.19 | 2.04 | -1.45 | 2.32 | 131.41 | < .01 |

This modification can be considered reasonable because, in accordance with what has been described in previous studies into flow [8, 4, 9, 10, 14], it was decided to propose a second-order factor model made up of one higher-order factor (Model 3). The goodness-of-fit indices of Model 3 were: $\chi^2$ (*d.f.* = 231; *p < .001*) = 340.54; RMSEA = .03 with an interval at 90% (.02 to .04); GFI = .99; and AGFI = .99; CFI = 1.00; TLI = .99; SRMR = .04; BIC = -280.07. As in Model 1 (six related factors), these results provide empirical support for the structure proposed (six factors explained by one second order factor) (see Table 2). Given these results, it could be considered that Models 1 (six related factors) and 3 (second order factor) present the same goodness of fit. Therefore, both of them could be used. The decision regarding which one should be used just depends on theoretical background.

The completely standardised solution of the structural model is shown in Fig 1.

**Table 2. Global goodness-of-fit indices of the four models.**

| | $\chi^2$ | d.f. | P | RMSEA | GFI | AGFI | CFI | TLI | SRMR | BIC |
|---|---|---|---|---|---|---|---|---|---|---|
| Model 1 | 341.46 | 237 | < .01 | .04 | .99 | .99 | 1.00 | 1.00 | .04 | -295.27 |
| Model 2 | 1292.39 | 238 | < .01 | .10 | .92 | .90 | .96 | .95 | .14 | 656.97 |
| Model 3 | 340.54 | 231 | < .01 | .03 | .99 | .99 | 1.00 | 1.00 | .04 | -280.07 |
| Model 4 | 304.19 | 228 | < .01 | .03 | .99 | .99 | 1.00 | 1.00 | .05 | -308.36 |

Model 1 = Six related factors model; Model 2 = five first-order factors explained by one second-order factor except for time; Model 3 = six first-order factors explained by one second-order factor; Model 4 = bifactor model.

**Table 3. Standardized solution for Model 1 (M1) and Bifactor Model (BM). The last column shows the loading in the general factor of the bifactor model (B).**

| Item | Merging | | Concentra. | | Control | | Conscious. | | Time | | Autotelic | | B |
|---|---|---|---|---|---|---|---|---|---|---|---|---|---|
| | M1 | BM | M1 | BM | M1 | BM | M1 | BM | M1 | BM | M1 | BM | |
| 1 | .70* | .26* | | | | | | | | | | | .53* |
| 7 | .74* | .52* | | | | | | | | | | | .65* |
| 13 | .74* | .75* | | | | | | | | | | | .81* |
| 19 | .86* | .66* | | | | | | | | | | | .55* |
| 2 | | | .82* | .62* | | | | | | | | | .03 |
| 8 | | | .82* | .23* | | | | | | | | | .80* |
| 14 | | | .93* | .57* | | | | | | | | | .51* |
| 20 | | | .92* | .58* | | | | | | | | | .72* |
| 3 | | | | | .84* | .27 | | | | | | | .88* |
| 9 | | | | | .90* | .10 | | | | | | | .59* |
| 15 | | | | | .91* | .47 | | | | | | | .14* |
| 21 | | | | | .91* | .10 | | | | | | | .79* |
| 4 | | | | | | | .84* | .67* | | | | | .49* |
| 10 | | | | | | | .91* | .72* | | | | | .76* |
| 16 | | | | | | | .88* | .67* | | | | | .88* |
| 22 | | | | | | | .96* | .67* | | | | | .58* |
| 5 | | | | | | | | | .74* | .84* | | | .21* |
| 11 | | | | | | | | | .88* | .89* | | | .81* |
| 17 | | | | | | | | | .98* | .93* | | | .59* |
| 23 | | | | | | | | | .97* | .87* | | | .75* |
| 6 | | | | | | | | | | | .92* | .40* | .89* |
| 12 | | | | | | | | | | | .91* | .43* | .64* |
| 18 | | | | | | | | | | | .94* | .45* | .23* |
| 24 | | | | | | | | | | | .92* | .59* | .78* |

Concentra. = Concentration; Conscious. = Consciousness.

* $p < .05$

Nevertheless, given the apparent contradictory results regarding the fit and definition of *time* in the flow state dimension, a bifactor model was tested in order to obtain evidence as to whether the items of *time* could be considered in the same way as the items of the other dimensions, or if conversely, these items have complementary hues related to flow. The goodness-of-fit indices of Model 4 were: $\chi^2$ (*d.f.* = 228; *p < .001*) = 304.19; RMSEA = .03 with an interval at 90% (.02 to .04); GFI = .99; and AGFI = .99; CFI = 1.00; TLI = 1.00; SRMR = .05; BIC = -308.36

**Table 4. Correlations between factors.**

| | merging | concentration | control | consciousness | time | autotelic |
|---|---|---|---|---|---|---|
| merging | 1.00 | | | | | |
| concentration | .50* | 1.00 | | | | |
| control | .67* | .83* | 1.00 | | | |
| consciousness | .55* | .49* | .67* | 1.00 | | |
| time | .27* | .15* | .06 | .06 | 1.00 | |
| autotelic | .59* | .76* | .82* | .54* | .23* | 1.00 |

* $p < .05$

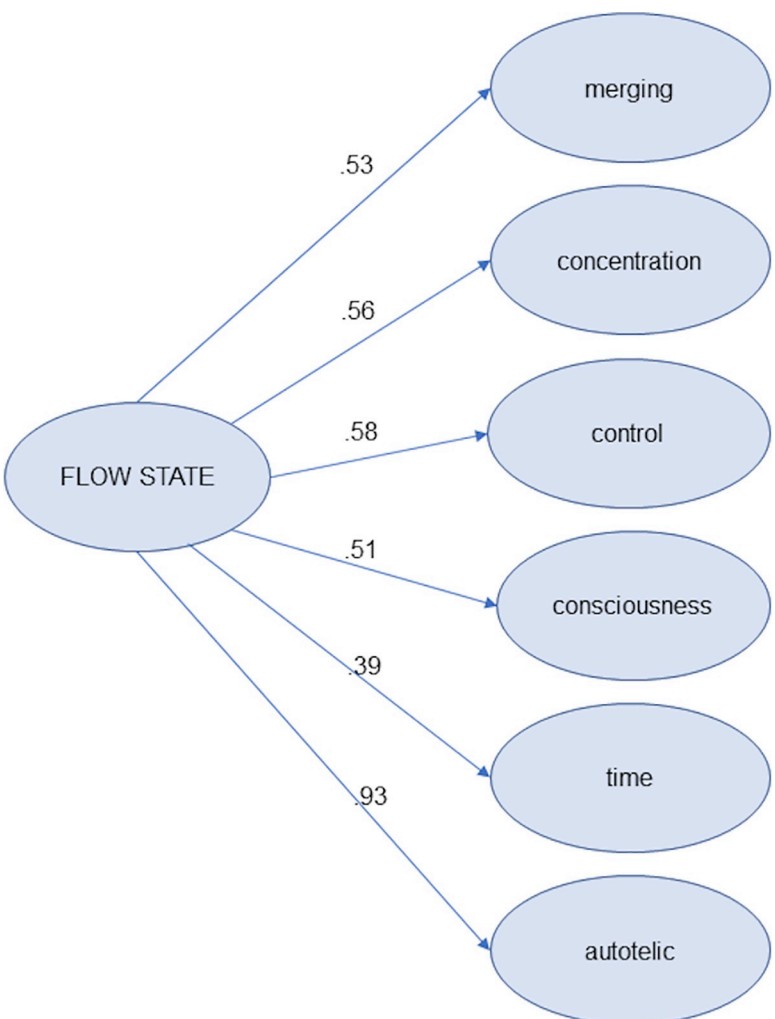

**Fig 1. Completely standardised solution of the structural model for Model 3.**

The standardised solution is shown in the last column of Table 3. As we can see, for *consciousness*, and especially for *time*, the items tend to show higher loadings in its original dimensions. This result is consistent with Model 3 (second order factor) where both dimensions are the most poorly predicted by flow state (see Fig 1). On the other hand, the loadings show that practically all the items reflect both a general factor and a specific factor, with the exception of Items 2, 9 and 21 (see Table 3).

## Reliability

The basic psychometric characteristics of the dimensions obtained in the CFA of Model 1 show that both the reliability of the scales and their average discrimination are adequate (Table 5). All six scales have Cronbach's alpha values above .80, the weighted construct omega presents values above .89 [49, 50] and their average discrimination is greater than .30 [51].

## Criterion validity

In order to analyse the criterion validity, we calculated the Pearson bivariate correlation index between each of the six subscales, the global flow, and the scores on: a) *balance*, b) *goals*, c)

**Table 5. Cronbach's α, omega, and mean discrimination.**

|  | Reliability (Cronbach's alpha) | Weighted Construct $\Omega_\omega$ | Mean discrimination |
|---|---|---|---|
| *Merging* | .81 | .86 | .64 |
| *concentration* | .89 | .94 | .79 |
| *Control* | .94 | .94 | .86 |
| *consciousness* | .93 | .97 | .84 |
| *Time* | .92 | .97 | .82 |
| *autotelic* | .95 | .95 | .88 |
| *FLOW* | .92 | .99 | .58 |

*feedback*. Given the inter-relation between these scales, we also obtained the partial correlation between each subscale, global flow included, and each one of the three measures referred to above, controlling in each case the effect of the two remaining scales that are conditions for entering flow (Table 6).

In the measurements of *balance*, *goals* and *feedback*, the correlations obtained with the 6 dimensions and global flow obtained a significance level of $p < .01$. This is except for the *time* scale with *goals* and *feedback* which are not significant. However, once the influence of the remaining subscales on each one of the dimensions had been controlled by means of partial correlations, the relationship between *goals* and all dimensions disappeared, *global flow* included, so that correlations ceased to be significant and close to 0 (Table 6). In *feedback*, the relationship is maintained with all dimensions and *global flow*, except with *time*, but with a slight decrease as well as a significance level of p < .05 on *merging* (pr = .10). Instead, the relationship with *time* (pr = -.09) is reversed and is significant to p < .05. Finally, the Pearson correlations obtained between *balance* and all dimensions, *global flow* included, in the partial correlations the relationship was maintained but there was a slight decrease in the direct relationship, except with *time* (pr = .19), which was slightly greater (Table 6).

## Discussion

This study has examined the factorial structure of the EFIM with a sample of Spanish musicians. As Calvo et al. [13] discussed in their work, the translation of this tool into Spanish and its adaptation can provide a very fruitful line of work and research. Moreover, according to Csikszentmihalyi [52], flow appears in all areas of life and is closely related to the satisfaction

**Table 6. Pearson correlations (r) and partial correlations (pr) between the dimensions of the EFIM and the criteria measurements (scales: balance, goals and feedback).**

|  | balance | | goals | | feedback | |
|---|---|---|---|---|---|---|
|  | *r* | *pr* | *r* | *pr* | *r* | *pr* |
| *Merging* | .53** | .35** | .39** | -.02 | .43** | .10* |
| *Concentration* | .58** | .21** | .58** | .04 | .67** | .36** |
| *Control* | .69** | .35** | .65** | -.04 | .79** | .52** |
| *consciousness* | .39** | .14** | .36** | -.03 | .44** | .22** |
| *Time* | .16** | .19** | .05 | -.01 | .02 | -.09* |
| *autotelic* | .61** | .39** | .47** | -.08 | .56** | .24** |
| *Global flow* | .69** | .42** | .58** | .04 | .66** | .32** |

\* *p* < .05

\*\* *p* < .01

of the activities we do. Therefore, this instrument can be modified for use in various areas of psychology, such as the psychology of music. In addition, we consider that the flow variable may be highly significant in psychophysiological processes that affect musicians in general and professional musicians in particular. It is not always easy for professional musicians to achieve a good level of performance, at the same time as personal satisfaction and well-being. Therefore, we believe that taking this variable into account in musicians would allow us to provide useful information for its application and for future research and interventions in musicians.

In this research, a structure of six factors that correspond to the scales of *merging*, *concentration*, *control*, *consciousness*, *time* and *autotelic* was analysed. This is in line with work by other authors [18] who have used these scales to measure the flow variable, specifically, an alignment of the item with the highest factor loading in the original studies [4]. These scales are defined as the six core components of the flow experience. The *balance*, *goals* and *feedback* dimensions are considered preconditions for flow and inherent to the task [25, 26] and in this research we have taken them into account as measurements for criteria validity.

The results of this study are like those of other research projects [8, 4, 9, 10, 14] in many ways. The six scales and global flow present good internal consistency and discrimination, and the structural Models 1 and 3 show good goodness-of-fit indices: Model 1 for a model of six related first-order factors and Model 3 for a hierarchical model of six first-order and one second-order factor. Therefore, either of the two models can be used to measure flow state. However, it should be considered that Model 3 is more in line with the theoretical framework, since, theoretically, the factors that make up the instrument are the elements that constitute the flow state [2]. In this way, the importance of each of the factors in the global flow state can be quantified, which can provide valuable information about their influence in achieving an optimal psychophysiological state [13].

The results show how *time*, despite its weak relationship with the rest of the dimensions (results which are in keeping with previous research [8, 4, 9, 10, 14]), is explained by a general flow factor (Model 3). Despite this weak relationship, we can consider that *time* is related to *merging* and *autotelic*, which would suggest it is part of the flow experience, but more related to the gratification of the experience and the fusion of action and thought, than with the other three dimensions, with which it may even maintain an inverse relationship that is hidden in bivariate correlations and apparently diminishes its relationship with both *merging* and *autotelic* and with global flow. According to Stavrou and Zervas [11], it is possible that global flow is actually multidimensional. The results suggest that *merging*, *time* and *autotelic* could fall within the same category, in which one perceives the most sensory and emotional part of the experience. An experience which, in addition, would be less controllable by the musician since it is a consequence of the state in which they find themselves. Meanwhile, *concentration*, *control* and *consciousness* could belong to another category which involves an assessment of cognitive aspects. It is possible that people are more familiar with thinking about them and, as the questionnaire states, in the statements regarding these three dimensions, respondents can respond not only as to whether they felt they were concentrating, in control of what they were doing and not worrying about the comments of others; they can also respond as to the attitude they adopted before playing, either because of their volition to do so or because they already rehearse putting into operation those cognitive mechanisms of concentration, thought control or attention to movement. These aspects are thus more controllable by the person. This could also help explain that *time* is a central component of the experience of flow as a sensation in different activities, although it is possible, as previous studies in sports psychology discuss [8, 4, 5, 9, 10], that is not thus either in all activities, or in all cases equally, due to factors such as the demands of the activity or personal characteristics. As Jackson and Eklund [5] comment,

future research may determine whether, or not, *time* depends on certain situations, types of activities or types of individuals.

In this research, a bifactor model was also analysed of which the goodness-of-fit indices are similar to those of Models 1 and 3. In addition, it allows us to observe how *time* reflects the general flow factor, while the high factor loading in its dimension reflect a specific part of that factor itself. This result is interesting given that, on the one hand, it supports the theoretical framework that considers *time* as an essential component of the experience of flow, despite obtaining, in general, low correlations with the other dimensions. On the other hand, it also gives rise to the need, as we mentioned above, to clarify what that specific contribution would consist of. The analysis of the bifactor model also shows that three items obtain low factor loadings. On the one hand, Items 9 and 21 obtain a loading of .10 in their *control* factor, which suggests that they are better explained by the general flow factor and not so much by its specific dimension. In fact, *control* presents only Item 15 with a relatively high loading. If we analyse how the *control* items are worded (https://www.mindgarden.com/100-flow-scales# horizontalTab2), we observe that only Item 15: *"I had a feeling of total control"*, differs from 3, 9 and 21, in that feeling of control is not focused on a specific matter, while the other items detail that the person felt or had a sensation of control over what he or she was doing or, specifically, over his or her own body. This difference as regards specifying over what they have control may explain why Item 15 has a factor loading that reflects that it is also explained by a specific factor and, meanwhile, the other three items seem to be better explained by the general flow factor, which would suggest the possibility of modifying and analysing them in order to explain the significance of these items in the *control* factor. On the other hand, Item 2, *concentration*, obtains a low load in the general flow factor, of .03 to be precise. One possible explanation is that the wording of Item 2: *"My attention was completely focused on what I was doing"*, unlike the other items on the *concentration* scale, speaks of focused attention and not specifically of concentration or of keeping one's mind on the specific task. Although the difference is very subtle, it may suggest that, at the specific moment of performance, people can sense the level of concentration for that task and, at the same time, that there are other moments, before, during or just after the performance, where the focus is on different stimuli and that this is not related to the state of flow during the performance.

It is also interesting to note how *goals* lose their relationship with the dimensions of the questionnaire once *feedback* and *balance* are controlled in the partial correlations. This result suggests that the clear goals in order to enter flow, which are important in the field of sport, where athletes set objectives and goals such as revalidating markers or improving them, are not the same type of objectives musicians set themselves. Taking into account that the items of the questionnaire were drawn up in relation to the field of sport, the sample of musicians may not identify with the statements as they are expressed in the questionnaire. This does not mean that musicians do not have clear goals because they do, both as regards the activity itself, and during a performance or during rehearsals and the preparation of a work, but it is probably not reflected in the scale of *goals* in such a way that musicians view it in that sense, as it could be with an affirmation of the type: I have a clear idea of how I am going to interpret the work, or, I have prepared the work with a clear technical-interpretative idea, for example. Beyond this possibility, there may also be differences as regards the contribution of *goals* to flow due to the focus of musical activity in our culture. It may be reflective of how in our country musical activity is not planned and musicians do not set goals in the same way that an Anglo-Saxon musician would do.

Among the limitations of the research it is worth mentioning that the snowball sampling technique, since it is not probabilistic, cannot guarantee that it is a representative sample of each one of the regions of Spain. However, it was possible to ensure that the selection of the

initial persons to whom the form was provided were from different regions of Spain and from different fields of music. Moreover, because it was launched via the social networks of the UNED, including those related to the field of music, the potential number of participants that could be reached increased. Another more specific limitation of this study is that, although it is considered more appropriate to complete the flow scale right at the end of the activity or task [2], the musicians were asked to answer the questions using the last musical activity in which they had participated as the "key situation", given the fact that musicians may not be available right after a performance or other type of musical event. However, this could enable us to generalise the factorial structure, in our case, of the EFIM in environments where respondents are not available after a performance [8], in this sense in order to obtain more validity evidences and be able to generalised the model tested into different groups (gender, instruments used, years of experience,. . .) an invariance analysis should be explore in forthcoming researches.

Finally, this study provides a validated tool to assess the state of flow in musicians. The validation of this instrument may have clinical and educational implications, since the use of the questionnaire allows one to identify significant aspects of what facilitates or inhibits a musical performance or of the learning itself. It can also be used for future research where researchers wish to measure the flow variable. However, the results of this study also suggest that flow can be explored in greater depth in musicians. Taking into account that the writing of the original items was based on the experience of athletes, for future research it would be more appropriate to initiate a line with a quantitative as well as qualitative approach to capture the flow experience of musicians in all its breadth: how they live their experiences and how they describe them. This could lead to a rewording of some items of the scales and another study of both the relationship between the criteria measurements and the dimensions of the questionnaire as possible alternative models.

## Supporting information

**S1 File. Polychoric correlations matrix.**
(PDF)

## Author Contributions

**Conceptualization:** Laura Moral-Bofill, Andrés Lópezdelallave, Mª. Carmen Pérez-Llantada.

**Data curation:** Laura Moral-Bofill, Francisco Pablo Holgado-Tello.

**Formal analysis:** Laura Moral-Bofill, Francisco Pablo Holgado-Tello.

**Investigation:** Laura Moral-Bofill, Andrés Lópezdelallave, Mª. Carmen Pérez-Llantada.

**Methodology:** Laura Moral-Bofill, Andrés Lópezdelallave, Mª. Carmen Pérez-Llantada, Francisco Pablo Holgado-Tello.

**Project administration:** Andrés Lópezdelallave, Mª. Carmen Pérez-Llantada.

**Supervision:** Andrés Lópezdelallave, Mª. Carmen Pérez-Llantada.

**Validation:** Laura Moral-Bofill, Andrés Lópezdelallave, Mª. Carmen Pérez-Llantada, Francisco Pablo Holgado-Tello.

**Visualization:** Laura Moral-Bofill, Andrés Lópezdelallave, Mª. Carmen Pérez-Llantada, Francisco Pablo Holgado-Tello.

**Writing – original draft:** Laura Moral-Bofill, Andrés Lópezdelallave, Mª. Carmen Pérez-Llantada.

**Writing – review & editing:** Laura Moral-Bofill, Andrés Lópezdelallave, Mª. Carmen Pérez-Llantada, Francisco Pablo Holgado-Tello.

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
