## [Decision Letter · Decision Letter 0]

27 Dec 2019

PONE-D-19-31155

Adaptation to Spanish and psychometric study of the Flow State Scale-2 in the field of musical performers

PLOS ONE

Dear DR. Fco. Pablo Holgado,

Thank you for submitting your manuscript to PLOS ONE. After careful consideration, we feel that it has merit but does not fully meet PLOS ONE’s publication criteria as it currently stands. Therefore, we invite you to submit a revised version of the manuscript that addresses the points raised during the review process.

We would appreciate receiving your revised manuscript by Feb 10 2020 11:59PM. To enhance the reproducibility of your results, we recommend that if applicable you deposit your laboratory protocols in protocols.io, where a protocol can be assigned its own identifier (DOI) such that it can be cited independently in the future. For instructions see: http://journals.plos.org/plosone/s/submission-guidelines#loc-laboratory-protocols

We look forward to receiving your revised manuscript.

Kind regards,

Eduardo Fonseca-Pedrero, PhD

Academic Editor

PLOS ONE

Additional Editor Comments:

The work entitled “Adaptation to Spanish and psychometric study of the Flow State Scale-2 in the field ofm usical performers” is of great interest in the assessment field. The research is very stimulating. It contains new scientific knowledge and provides comprehensive information for further development in this productive line of research. This research is well-argued and clearly worthy of publication. As minor comments I would like to say:

0.- Add empirical results in the abstract.

1.- Add relevant references to understand psychometric procedures used in this research: Muñiz J, Fonseca-Pedrero E. Ten steps for test development. Psicothema. 2019 Feb;31(1):7-16.

2.- Add the main goals and hypotheses at the end of the intro.

3.- Add more information about sampling procedure and sample characteristics. In particular, add more information about the all sociodemographic characteristics of the sample, e.g., ethnicity, socio-economic status, etc. Here we have convenience sample, and we know the limitations related to this kind of samples.

4.-Do you have any information about non-response? Describe inclusion/exclusion criteria if part of the data was excluded from the analysis. Were outliers removed from the data? Please, Which method did you use to deal with missing data in the analyses? What variables are related to missing data?

5.- Please, add subsections in Results (descriptive, CFA, reliability).

6.- Please, test other CFA models, e.g., bifactor model, unidimensional model, etc.

7.-Please, add other CFA goodness of fit indices (e.g., BIC, TLI).

8.-Please explain the constrains of the present study (e.g., not use infrequency or social desirability scale, sample and sampling, self-reports, etc.).

9.-Please, delete from table 1, N=486.

10.- Add as supplementary material, the polychoric correlations matrix.

11.- Please add information from the ethical committee.

12.- Please, following the PLOS ONES standards.

Journal Requirements:

2. We noticed you have some minor occurrence of overlapping text with the following previous publication, which needs to be addressed:

Ashinoff, Brandon K., and Ahmad Abu-Akel. "Hyperfocus: the forgotten frontier of attention." Psychological research (2019): 1-19.

In your revision ensure you cite all your sources (including your own works), and quote or rephrase any duplicated text outside the methods section. Further consideration is dependent on these concerns being addressed.

3. Please include a copy in the original language of the adapted questionnaire you developed, as Supporting Information.

Reviewers' comments:

Reviewer's Responses to Questions

**Comments to the Author**

1. Is the manuscript technically sound, and do the data support the conclusions?

Reviewer #1: Yes

Reviewer #2: Yes

2. Has the statistical analysis been performed appropriately and rigorously? 

Reviewer #1: Yes

Reviewer #2: Yes

3. Have the authors made all data underlying the findings in their manuscript fully available?

Reviewer #1: Yes

Reviewer #2: Yes

4. Is the manuscript presented in an intelligible fashion and written in standard English?

Reviewer #1: Yes

Reviewer #2: Yes

5. Review Comments to the Author

Reviewer #1: The paper Adaptation to Spanish and psychometric study of the Flow State Scale-2 in the field of musical performers aims to provide validity evidence for the Spanish adaptation of the Flow State Scale-2 (FSS-2) in a sample of musicians. The adapted scale is called Estado de Flujo para Intérpretes Musicales (EFIM). The original FSS-2 scale has 9 factors whereas the Spanish version (EFIM) is shorter, containing 6 of the 9 original factors.

I think the manuscript is good, it is well structured and well written. I enjoyed reading it, and for that, I would like to congratulate the authors. The study meets the expected conditions for a piece of research and I recommend its publication. I do, however, have a few minor questions that I believe could help to improve the paper. I would like to hear the authors’ responses.

- The authors state that the EFIM scale contains the six factors that “provide the highest factorial loads in the original studies” (referring to FSS-2). However, this is not entirely accurate: in the study by Jackson & Eklund (2002) neither Time nor Consciousness were among the six factors with the highest factorial loads. Instead, Goals and Balance were among those six factors, but were not included in the EFIM. Therefore believe the authors should redefine the reasons for choosing the six factors for the Spanish adaptation.

- In terms of how the 9 original factors of the FSS-2 were managed, I have another question. In another part of the paper, the authors indicate that the three factors from the FFS-2 not included in the EFIM (Feedback, Goals and Balance) “are considered preconditions for Flow” (although I’m not sure about this statement). They use this to justify the use of these three factors in the predictive validity section. However, isn’t it a little opportune to take a part (in this case the three segregated factors of the FSS-2) to justify the whole (in this case the EFIM)? Imagine if we validated a short version of the verbal factor of the WISC-R (for example, with the subscales Information, Similarities and Comprehension) and to demonstrate the predictive validity of these three scales we correlate them with the Vocabulary scale of the original test? Aren’t the authors doing the same here? Given that it seems as though the authors have information for the nine original scales of the FSS-2, have they tried to fit a 9-factor model? What were the results? Even if they do not give this information in the published manuscript, I would very much like to read their response, and if possible, see the fit data for the nine-factor model.

- One question about the wording. In lines 307-311, the authors state that “In order to analyse the criterion validity we calculated the Pearson bivariate correlation index between each of the six subscales, the global flow, and the scores on: a) balance, b) goals, c) feedback. Given the inter-relation between these scales, we also obtained the partial correlation between each subscale, global flow included, and each one of the three measures referred to above, controlling in each case the effect of the remaining scales” which seems to indicate that they analysed the effect between each pair of variables, discounting the effect of the other subscales. However, later (lines 395-396) they state “It is also interesting to note how goals lose their relationship with the dimensions of the questionnaire once feedback and balance are controlled in the partial correlations”. It is not clear which variables they used as controls to estimate the partial correlations.

- I am Spanish, and this comment comes from me thinking if I had to find the EFIM in a Spanish Psychology Test library. My history of classical philosophy professor said that we should require a name (whether a journal or a test etc.) to “determine and explain the content” [González Escudero, S. (1989). A propósito del nombre. Psicothema, 1(1), 5-6]. I’m afraid that the translation “Estado de Flujo para Intérpretes Musicales” does not do this. I think that in the context of the test the term flow could be translated into Spanish as : fluidez, plenitud, entusiasmo, compleción, enardecimiento or maybe even effluxion (which is at least the most similar to the original English, although there is the risk of a second meaning in Spanish: “un mal parto” a bad birth). Would it be better to make a “less literal” translation of the term Flow for potential Spanish users?

- In the references, there are some errors: the separation of words is incorrect in some cases (for example TheSpanishJournal of Psychology), there are inconsistencies when using journal names (for example: ,: Journal of Sport andExercise Psychology (sic) and Journal of Sport & Exercise Psychology), journals that are not capitalized (Frontiers in psychology, Experimental aging research), poorly referenced authors (for example, ¿Lopez SJ or Lopez JS?), variations in citing books or manuals in the order “City: publisher”. There are various other errors. Similarly, there are various studies that do not have a DOI where one is available. Please review all of the references: a poorly cited reference will not appear in search engines or impact indexers later.

- I note that there is a lot of information about the construction of Likert scales. Here are three (APA style) references that I am sure will interest the authors and may be included in their paper:

* Calderón, C., Navarro, D., Lorenzo-Seva, U., & Ferrando, P. J. (2019). Multidimensional or essentially unidimensional? A multi-faceted factor analytic approach for assessing the dimensionality of tests and items. Psicothema, 31(3), 450-457. https://doi.org/10.7334/psicothema2019.153

* Suárez-Alvarez, J., Pedrosa, I., Lozano, L. M., García-Cueto, E., Cuesta. M., & Muñiz, J. (2018) Using reversed items in Likert scales: A questionable practice. Psicothema, 30(2), 149-158. doi: 10.7334/psicothema2018.33

* Lozano, L. M., García-Cueto, E., & Muñiz, J. (2008). Effect of the number of response categories on the reliability and validity of rating scales. Methodology, 4(2), 73-79. doi:10.1027/1614-2241.4.2.73

In summary, it is very good work methodologically. The small, formal defects and issues about evidence of validity can, I believe, be corrected without too much trouble by the authors, and the manuscript subsequently published in PLOS-One.

Reviewer #2: I would like to thank the authors for their submission to PLOS ONE. I enjoyed reading this excellent article and I only have a few minor suggestions to make. My comments are below:

- In the Results section (Confirmatory Factor Analysis) I would recommend including a summary table that gives the goodness of fit indices X2, RMSEA, GFI, AGFI, NFI and SRMR regarding the models compared in the text: model 1 (266); model 2 (280) and model 3 (288). This table would help the reader to compare the models and reinforce the arguments in favor of selecting the model with the best fit.

- In the light of those goodness of fit indicators, it is interesting because the removal of time as a constituent factor of the six-factor model 1 results in a new, five-factor model 2 with indicators suggesting a worse fit (in GFI, AGFI and in NFI). Although the authors do comment on this in the discussion section, from a psychometric point of view it is counterintuitive that on removing the factor that clearly provides less to the solution, the fit of the model is substantially worse. In the end, what role, in psychometric terms, does the factor Time play? Might it not be understood as an inconsistency or a certain amount of instability in the construct the authors want to validate?

- In the discussion (349-355) the authors explain their reasons for using a six-factor scale that excludes the factors balance, goals and feedback rather than the nine-factor scale produced for the original study, according to Jackson and Eklund, 2002. They state that the six remaining factors are those with the highest factorial loadings, although the cited study indicates that the three excludable factors due to the lowest loadings were in fact time, consciousness and feedback. The authors should consider reframing their argument for choosing to exclude balance and goals.

-Please consider including the DOI in the references wherever it is available, for example, in the following:

14. Wrigley WJ, Emmerson SB. The experience of the flow state in live music performance. Psychology of Music. 2013;41: 292-305. DOI: 10.1177/0305735611425903

16. Custodero LA. Seeking challenge, finding skill: Flow experience and music education. ArtsEducationPolicy Review. 2002;103: 3–9. https://doi.org/10.1080/10632910209600288

18. Fullagar CJ, Knight PA, Sovern HS. Challenge/skill balance, flow, and

performance anxiety. Applied Psychology. 2013;62: 236-259. DOI:10.1111/j.1464-0597.2012.00494.x

20. Srinivasan N, Gingras B. Emotional intelligence predicts individual differences in proneness for flow among musicians: the role of control and distributed attention. Frontiers in psychology. 2014;5: 608 https://doi.org/10.3389/fpsyg.2014.00608

37. Hambleton RK, Merenda P, Spielberger C. Adapting educational and psychological tests for cross-cultural assessment. Hillsdale: Lawrence Erlbaum Publishers; 2005. DOI: 10.1007/S11336-007-9014-3

42. Holgado FP, Chacón S, Barbero I, Vila, E. Polychoric 553 versus Pearson correlations 554 in exploratory and confirmatory factor analysis of ordinal variables. Quality & 555 Quantity. 2010;44: 153-166. https://doi.org/10.1007/s11135-008-9190-y

46. Yanuar F, Devianto D, Marisa S, Zetra, A. Consistency test of reliability index in SEM model. Applied Mathematical Sciences. 2015;9: 5283-5292 http://dx.doi.org/10.12988/ams.2015.56446.

In general, this article is in an excellent state and, with a few additions and clarifications, it will be a significant contribution to the literature on flow. The authors should be thanked for their excellent work in this valuable project.

6. PLOS authors have the option to publish the peer review history of their article (what does this mean?). If published, this will include your full peer review and any attached files.

Reviewer #1: No

Reviewer #2: No

---

## [Author Response · Author response to Decision Letter 0]

30 Jan 2020

Dear Dr. Eduardo Fonseca-Pedrero,

First, we would like to thank you all, the comments that you have done so kindly about our work. Sincerely, we believe that with your comments, this work has improved in its results and, even, they open more questions which we believe can be further investigated. 

Next, you can read our responses about all the comments that you have asked us. We hope the responses are to your liking. 

Thank you very much.

We look forward to hearing your feedback.

Editor Comments:

0.- Add empirical results in the abstract. 

Finally, we have rewritten the summary in this way:

Flow is a positive and optimal state of mind, during which people are highly motivated and absorbed in the activity they are doing. It is an experience that can occur in any area of life. One of the measurement instruments which is most commonly used to evaluate this construct is the Flow State Scale-2 (FSS-2). This instrument has been used in different languages, mainly in the field of sport. In this research work, the FSS-2 has been translated into Spanish and administered to 486 musicians from different regions of Spain in order to examine the psychometric properties. A version which uses six dimensions from the original questionnaire has been used - those that constitute the experience of flow - and four alternative models have been analysed (Six factor model, two second order factor models and a bifactorial model). The results revealed that the dimension time could be controversial and more research could be needed. In general terms, the six-factor model and a second-factor one are solutions consistent with previous studies and show that the questionnaire can be considered a reliable and useful tool, both in clinical and educational contexts, as well as an instrument for the evaluation of this construct in future research. However, the results of this study also suggest that flow can be explored in greater depth in musicians, taking into account that the writing of the original items was based on the experience of athletes and that the role of time in flow needs to be investigated. 

1.- Add relevant references to understand psychometric procedures used in this research: Muñiz J, Fonseca-Pedrero E. Ten steps for test development. Psicothema. 2019 Feb;31(1):7-16. 

The new reference has been included in the following paragraph:

“In order to obtain evidence of construct validity of the instrument in a Spanish sample, we tested the original model proposed by Jackson and Eklund [4] using the Confirmatory Factor Analysis (CFA) procedure [39].”

2.- Add the main goals and hypotheses at the end of the intro. 

The following paragraph has been added in the paper:

On the one hand, it was hypothesised that the results of the psychometric analysis of the FSS-2 scale adapted and translated into Spanish in a population of performing musicians would be consistent with the results obtained in other studies that have examined the psychometric properties of the FSS-2. On the other hand, a second hypothesis was presented according to which, if the factors of skill-challenge balance, clear objectives and clear feedback are conditions in order to enter the flow state, they will positively correlate with the other factors that represent the flow experience or state. This correlation can therefore be considered as supporting the criterion validity.

3.- Add more information about sampling procedure and sample characteristics. In particular, add more information about the all sociodemographic characteristics of the sample, e.g., ethnicity, socio-economic status, etc. Here we have convenience sample, and we know the limitations related to this kind of samples. 4.- Do you have any information about non-response? Describe inclusion/exclusion criteria if part of the data was excluded from the analysis. Were outliers removed from the data? Please, Which method did you use to deal with missing data in the analyses? What variables are related to missing data? 

The following information relating to the issues you raised has been added in the paper:

A sample of 558 participants was obtained, including music students as well as amateurs and professionals. They were all Spanish speakers and came from different regions of Spain. As a criterion for inclusion, it was established that participants would have a well-established relationship with musical performance (students, professionals, amateurs), specifically at least two years of study; as a criterion for exclusion a minimum age of 18 was established. As a consequence of these criteria, 72 subjects were excluded.

The form was published via the Google Forms tool, in which the EFIM questionnaire was included. The questions were organised in such a way that it was “mandatory” to answer all of them (Google signals this requirement with a red asterisk at the end of each question). The participants thus answered all the questions in the survey and there were no cases where the answers to any of the questions set by the tool were missing. 

5.- Please, add subsections in Results (descriptive, CFA, reliability). 

Two new subsections have been included:

- Descriptive Statistics

- Reliability 

6.- Please, test other CFA models, e.g., bifactor model, unidimensional model, etc. 

We really appreciated this suggestion. We have conducted a bifactorial model, and in a way it helped to understand the previous results regarding the time dimension. We have included the following paragraph: 

Nevertheless, given the apparent contradictory results regarding the fit and definition of time in the flow state dimension, a bifactorial model was tested in order to obtain evidence as to whether the items of time could be considered in the same way as the other items of the other dimensions, or if conversely, these items have complementary hues related to flow. The goodness-of-fit indices of Model 3 were: �2 (d.f. = 231; p < .001) = 304.19; RMSEA = .028 with an interval at 90% (.019 to .036); GFI = .99; and AGFI = .99; CFI = 1.00; NFI = .99; SRMR = .05. The standardised solution is shown in the last column of Table 2. As we can see, for consciousness, and especially for time, the items tend to show higher loadings in its original dimensions. This result is consistent with Model 3 where both dimensions are the most poorly predicted by flow state. 

The results are discussed in the Discussion section.

7.-Please, add other CFA goodness of fit indices (e.g., BIC, TLI). 

TLI index also is named NNFI, that were reported. However, according to the reviewer

suggestion the name has been modified to TLI, that may be is more known.

8.-Please explain the constrains of the present study (e.g., not use infrequency or social desirability scale, sample and sampling, self-reports, etc.). 

Regarding the constraints of the study, the following comments have been added:

Among the limitations of the research it is worth mentioning that the snowball sampling technique, since it is not probabilistic, cannot guarantee that it is a representative sample of each and every one of the regions of Spain. However, it was possible to ensure that the selection of the initial subjects to whom the form was provided from different regions of Spain and from different fields of music. Moreover, because it was launched via the social networks of the UNED, including those related to the field of music, the potential number of subjects that could be reached increased. *was

9.-Please, delete from table 1, N=486. 

Okay, the column has been removed.

10.- Add as supplementary material, the polychoric correlations matrix.

We will submit the polychoric correlations matrix as Supporting Information.

11.- Please add information from the ethical committee. 

In the materials and methods section, we have added the following paragraph:

“The research has been carried out following the standards recommended for research on human subjects from the code of ethics of the European Community and the American Psychological Association´s Ethical Standards for Research and Publication. The research was approved by the Bioethics Committee of the UNED. We have guaranteed privacy in the processing of data. Participation in the study was voluntary and anonymous.”

Journal Requirements: 

2. We noticed you have some minor occurrence of overlapping text with the following previous publication, which needs to be addressed:

Ashinoff, Brandon K., and Ahmad Abu-Akel. "Hyperfocus: the forgotten frontier of attention." Psychological research (2019): 1-19.

Okay, we have taken note of the overlapping in that publication. We have worded it as follows:

Recently, flow theory has begun to distinguish between the conditions that are necessary to give rise to flow and the psychological components that constitute the experience of flow [25, 26]. Therefore, in order to enter flow, an appropriate balance between the skills and challenges that a person faces is deemed necessary, as well as having clear objectives which are proximate to the action and, also, that there is clear and immediate feedback. The other dimensions would be the subjective experience of the flow state: concentration on the task, action-awareness, sense of control, loss of self-consciousness, transformation of time and autotelic experience [25, 26].

3. Please include a copy in the original language of the adapted questionnaire you developed, as Supporting Information. 

Okay, we will submit it as Supporting Information

Reviewer #1: 

1.- The authors state that the EFIM scale contains the six factors that “provide the highest factorial loads in the original studies” (referring to FSS-2). However, this is not entirely accurate: in the study by Jackson & Eklund (2002) neither Time nor Consciousness were among the six factors with the highest factorial loads. Instead, Goals and Balance were among those six factors, but were not included in the EFIM. Therefore, believe the authors should redefine the reasons for choosing the six factors for the Spanish adaptation.

Thanks for the comment. It is clearly a mistake due to confusion during the writing and translation. The correct paragraph reads as follows:

In this research, a structure of six factors that correspond to the scales of merging, concentration, control, consciousness, time and autotelic was analysed. This is in line with work by other authors [18] who have used these scales to measure the flow variable, specifically, an alignment of the item with the highest factorial load in the original studies [4]. 

2.- In terms of how the 9 original factors of the FSS-2 were managed, I have another question. In another part of the paper, the authors indicate that the three factors from the FFS-2 not included in the EFIM (Feedback, Goals and Balance) “are considered preconditions for Flow” (although I’m not sure about this statement). They use this to justify the use of these three factors in the predictive validity section. However, isn’t it a little opportune to take a part (in this case the three segregated factors of the FSS-2) to justify the whole (in this case the EFIM)? Imagine if we validated a short version of the verbal factor of the WISC-R (for example, with the subscales Information, Similarities and Comprehension) and to demonstrate the predictive validity of these three scales we correlate them with the Vocabulary scale of the original test? Aren’t the authors doing the same here? Given that it seems as though the authors have information for the nine original scales of the FSS-2, have they tried to fit a 9-factor model? What were the results? Even if they do not give this information in the published manuscript, I would very much like to read their response, and if possible, see the fit data for the nine-factor model. 

The reviewer raises a significant theoretical question. In our research, we have chosen a model which is consistent with our theoretical hypothesis concerning the structure of flow.

We believe that this structure of six factors that correspond to the experience or state of flow and the three factors that are the conditions in order for that state to be achieved is well argued in the theoretical framework of the theory of flow by Nakamura and Csíkszentmihályi. The experience of flow, as a state a person finds themselves in and about which this person can speak, refers to the assessments about the internal state itself regarding the activity or performance in a task. These assessments are the result of the observation of one's own experience, which can be in the form of sensations, emotions or cognitive functioning. However, the skill-challenge balance, having objectives which are proximate to the action and immediate clear feedback, do not reflect the experience of flow itself as an internal state. Although the subjective belief of the balance between skills/challenges in order to enter a state of flow is significant, it is also important that this balance be objectively real, even if the person is not aware of this. This means that, for example, in learning processes, students can reach a state of flow, rather than a state of anxiety, if that balance is ensured. Furthermore, although the objectives which are proximate to the action entail that the person is clear about what he or she has to do, and, therefore, they are able to assess it, it is the result of that clarity that probably translates into a state of control and concentration, for example. With immediate feedback, something similar would happen. The sources of feedback can be diverse and also external to the person, and the FSS-2 includes the assessment made by the person as to how their performance is proceeding. It is therefore a reflection as to whether or not he or she is aware of what they are doing and how they are doing it, but the internal state would relate to the ability to keep their focus on those feedback elements.

Furthermore, research has been conducted where the dimensions that are deemed conditions to enter flow have been controlled, especially the dimension of skill-challenge balance in activities such as online games or learning situations.

Pearce, J. M., Ainley, M., & Howard, S. (2005). The ebb and flow of online learning. Computers in human behavior, 21(5), 745-771.

Rheinberg, F., & Vollmeyer, R. (2003). Flow-Erleben in einem Computerspiel unter experimentell variierten Bedingungen.

Keller, J., & Bless, H. (2008). Flow and regulatory compatibility: An experimental approach to the flow model of intrinsic motivation. Personality and social psychology bulletin, 34(2), 196-209.

Moller, A. C., Csikszentmihalyi, M., Nakamura, J., & Deci, E. L. (2007). February. Developing an experimental induction of flow. In Poster presented at the Society for Personality and Social Psychology Conference, Memphis, TN.

Moller, A. C., Meier, B. P., & Wall, R. D. (2010). Developing an experimental induction of flow: Effortless action in the lab. In B. Bruya (Ed.), Effortless attention: A new perspective in the cognitive science of attention and action (p. 191–204). MIT Press. https://doi.org/10.7551/mitpress/9780262013840.003.0010

Therefore, this theoretical definition of flow is what we have tried to investigate empirically by means of the CFA. Given the results obtained, we believe that our hypothesis was able to be empirically sustained.

However, below we provide the results for the model with nine dimensions: 

Degrees of Freedom = 558

Normal Theory Weighted Least Squares Chi-Square = 2298.33 (P = 0.0)

Satorra-Bentler Scaled Chi-Square = 879.57 (P = 0.00)

Estimated Non-Centrality Parameter (NCP) = 321.57

90 Percent Confidence Interval for NCP = (244.88; 406.18)

Minimum Fit Function Value = 1.42

Population Discrepancy Function Value (F0) = 0.74

90 Percent Confidence Interval for F0 = (0.56; 0.93)

Root Mean Square Error of Approximation (RMSEA) = 0.036

90 Percent Confidence Interval for RMSEA = (0.032; 0.041)

P-Value for Test of Close Fit (RMSEA < 0.05) = 1.00

Expected Cross-Validation Index (ECVI) = 2.52

90 Percent Confidence Interval for ECVI = (2.34; 2.71)

ECVI for Saturated Model = 3.06

ECVI for Independence Model = 148.44

Chi-Square for Independence Model with 630 Degrees of Freedom = 64498.87

Independence AIC = 64570.87

Model AIC = 1095.57

Saturated AIC = 1332.00

Independence CAIC = 64753.66

Model CAIC = 1643.95

Saturated CAIC = 4713.71

Normed Fit Index (NFI) = 0.99

Non-Normed Fit Index (NNFI) = 0.99

Parsimony Normed Fit Index (PNFI) = 0.87

Comparative Fit Index (CFI) = 0.99

Incremental Fit Index (IFI) = 0.99

Relative Fit Index (RFI) = 0.98

Critical N (CN) = 316.85

Root Mean Square Residual (RMR) = 0.046

Standardised RMR = 0.046

Goodness-of-Fit Index (GFI) = 0.99

Adjusted Goodness-of-Fit Index (AGFI) = 0.99

Parsimony Goodness-of-Fit Index (PGFI) = 0.83

While this model also presents appropriate fit indices, there are some issues to be considered:

1. We have tried to follow a theory driven perspective, and our interest is thus focused on the model presented in the paper.

2. If we compare this model with the one presented in the paper, the relative fit index based on Chi-squared test, is worse in the model with nine dimensions. On the other hand, it is a logical result because its complexity is higher. This model is also tenable but it is not consistent with our theoretical position presented in the paper. 

Nonetheless, if the reviewer considers that it could be of interest to present this result in the paper, please let us know. We felt that the paper was sufficient in length, and the presence of more results could hinder the understanding of the paper by potential readers with several results which do not relate to the main objective. 

3.- One question about the wording. In lines 307-311, the authors state that “In order to analyse the criterion validity we calculated the Pearson bivariate correlation index between each of the six subscales, the global flow, and the scores on: a) balance, b) goals, c) feedback. Given the inter-relation between these scales, we also obtained the partial correlation between each subscale, global flow included, and each one of the three measures referred to above, controlling in each case the effect of the remaining scales” which seems to indicate that they analysed the effect between each pair of variables, discounting the effect of the other subscales. However, later (lines 395-396) they state “It is also interesting to note how goals lose their relationship with the dimensions of the questionnaire once feedback and balance are controlled in the partial correlations”. It is not clear which variables they used as controls to estimate the partial correlations.

Thanks for the comment. It is indeed confusing. The paragraph could be clarified as follows:

Given the inter-relation between these scales we also obtained the partial correlation between each subscale, global flow included, and each one of the three measures referred to above, controlling in each case the effect of the two remaining scales that are conditions for entering flow (Table 6). 

4.- I am Spanish, and this comment comes from me thinking if I had to find the EFIM in a Spanish Psychology Test library. My history of classical philosophy professor said that we should require a name (whether a journal or a test etc.) to “determine and explain the content” [González Escudero, S. (1989). A propósito del nombre. Psicothema, 1(1), 5-6]. I’m afraid that the translation “Estado de Flujo para Intérpretes Musicales” does not do this. I think that in the context of the test the term flow could be translated into Spanish as : fluidez, plenitud, entusiasmo, compleción, enardecimiento or maybe even effluxion (which is at least the most similar to the original English, although there is the risk of a second meaning in Spanish: “un mal parto” a bad birth). Would it be better to make a “less literal” translation of the term Flow for potential Spanish users?

We really like your assessment of the term, and we find that, although there is a lot of literature - both academic and informative - concerning flow where it is translated as “estado de flujo”, the term "fluidez" or even "fluencia" is also used. We believe that with regard to musicians, translating it as "fluidez" may be more accurate because it conveys the quality of expressing oneself with a certain degree of ease and spontaneity. 

5.- In the references, there are some errors: the separation of words is incorrect in some cases (for example TheSpanishJournal of Psychology), there are inconsistencies when using journal names (for example: ,: Journal of Sport andExercise Psychology (sic) and Journal of Sport & Exercise Psychology), journals that are not capitalized (Frontiers in psychology, Experimental aging research), poorly referenced authors (for example, ¿Lopez SJ or Lopez JS?), variations in citing books or manuals in the order “City: publisher”. There are various other errors. Similarly, there are various studies that do not have a DOI where one is available. Please review all of the references: a poorly cited reference will not appear in search engines or impact indexers later.

We have reviewed the references.

6.- I note that there is a lot of information about the construction of Likert scales. Here are three (APA style) references that I am sure will interest the authors and may be included in their paper:

Thank you very much. We will add the following reference:

* Lozano, L. M., García-Cueto, E., & Muñiz, J. (2008). Effect of the number of response categories on the reliability and validity of rating scales. Methodology, 4(2), 73-79. doi:10.1027/1614-2241.4.2.73

Reviewer #2: 

1.- In the Results section (Confirmatory Factor Analysis) I would recommend including a summary table that gives the goodness of fit indices X2, RMSEA, GFI, AGFI, NFI and SRMR regarding the models compared in the text: model 1 (266); model 2 (280) and model 3 (288). This table would help the reader to compare the models and reinforce the arguments in favor of selecting the model with the best fit. 

Thanks for the suggestion. This summary table with the goodness-of-fit indices has been added to the paper:

Table 2. Global goodness-of-fit indices of the four models.

 �2 d.f. p RMSEA GFI AGFI CFI NFI SRMR

Model 1 341.46 237 <.001 .036 .99 .99 1.00 .99 .04

Model 2 1292.39 238 <.001 .10 .92 .90 .96 .95 .14

Model 3 340.54 231 <.001 .033 .99 .99 1.00 .99 .04

Model 4 304.19 228 <.001 .028 .99 .99 1.00 .99 .05

Model 1= Related six-factor model; Model 2= five first-order factors and one second-order factor (not time); Model 3= six first-order factors and one second-order factor; Model 4= bifactorial model.

2.- In the light of those goodness of fit indicators, it is interesting because the removal of time as a constituent factor of the six-factor model 1 results in a new, five-factor model 2 with indicators suggesting a worse fit (in GFI, AGFI and in NFI). Although the authors do comment on this in the discussion section, from a psychometric point of view it is counterintuitive that on removing the factor that clearly provides less to the solution, the fit of the model is substantially worse. In the end, what role, in psychometric terms, does the factor Time play? Might it not be understood as an inconsistency or a certain amount of instability in the construct the authors want to validate?

We have analysed a bifactorial model that we have included in the article and which we have commented on in the discussion section. The goodness-of-fit indices are similar to those of Models 1 and 3. However, it allows us to observe how time reflects the general flow factor, while the high factorial loads in its dimension reflect a specific part of that factor itself. This result, on the one hand, supports the theoretical framework that considers time as an essential component of the experience of flow, despite obtaining, in general, low correlations with the other dimensions. On the other hand, it also gives rise to the need, as we state in the article, to clarify what that specific contribution would consist of.

3.- In the discussion (349-355) the authors explain their reasons for using a six-factor scale that excludes the factors balance, goals and feedback rather than the nine-factor scale produced for the original study, according to Jackson and Eklund, 2002. They state that the six remaining factors are those with the highest factorial loadings, although the cited study indicates that the three excludable factors due to the lowest loadings were in fact time, consciousness and feedback. The authors should consider reframing their argument for choosing to exclude balance and goals.

Thank you for the comment. It is clearly a mistake due to confusion during the writing and translation. The corrected paragraph reads as follows:

In this research, a structure of six factors that correspond to the scales of merging, concentration, control, consciousness, time and autotelic has been analysed. This is in line with work by other authors [18] who have used these scales to measure the flow variable, specifically, an alignment of the item with the highest factorial load in the original studies [4].

4.- Please consider including the DOI in the references wherever it is available.

Thank you very much. We have included all the references that we have found.

Thank you very much.

Best regards,

Laura Moral-Bofill

Pablo Holgado Tello

---

## [Editor Report · Decision Letter 1]

26 Feb 2020

PONE-D-19-31155R1

Adaptation to Spanish and psychometric study of the Flow State Scale-2 in the field of musical performers

PLOS ONE

Dear Holgado-Tello,

Thank you for submitting your manuscript to PLOS ONE. After careful consideration, we feel that it has merit but does not fully meet PLOS ONE’s publication criteria as it currently stands. Therefore, we invite you to submit a revised version of the manuscript that addresses the points raised during the review process.

We would appreciate receiving your revised manuscript by 25-3-2020. To enhance the reproducibility of your results, we recommend that if applicable you deposit your laboratory protocols in protocols.io, where a protocol can be assigned its own identifier (DOI) such that it can be cited independently in the future. For instructions see: http://journals.plos.org/plosone/s/submission-guidelines#loc-laboratory-protocols

We look forward to receiving your revised manuscript.

Kind regards,

Eduardo Fonseca-Pedrero, PhD

Academic Editor

PLOS ONE

Additional Editor Comments (if provided):

1.- Add empirical information in the abstract (eg. reliability).

2.- APA style: better talk about persons, not subjects (see participants section).

3.- Used only two decimals (see method).

4.- Add SPSS reference.

5.- Please, explain all CFA models tested.

6.- Please test one factor model.

7.- Please check typo: bifactorial model (bifactor model).

8.- Table 3. Add p values.

9.- CFA: add BIC values and IC 90% RMSEA.

10.- Table 4, add p values.

11.- Please, add CFA measurement invariance analyses.

12.- Please, add information about the psychometric properties of: Challenge-skill balance (balance); Clear goals (goals); Unambiguous feedback (feedback).

---

## [Author Response · Author response to Decision Letter 1]

10 Mar 2020

Additional Editor Comments (if provided):

1.- Add empirical information in the abstract (eg. reliability). 

We have included some global fit indexes for the model of 6 related factor and the second order factor model. Also, the alphas range of the dimensions have been included. The new abstract is:

“Flow is a positive and optimal state of mind, during which people are highly motivated and absorbed in the activity they are doing. It is an experience that can occur in any area of life. One of the measurement instruments which is most commonly used to evaluate this construct is the Flow State Scale-2 (FSS-2). This instrument has been used in different languages, mainly in the field of sport. In this research work, the FSS-2 has been translated into Spanish and administered to 486 musicians from different regions of Spain in order to examine the psychometric properties. A version which uses six dimensions from the original questionnaire has been used - those that constitute the experience of flow - and four alternative models have been analysed (Six related factors model, two second order factor models and a bifactor model).The results revealed that the dimension time could be controversial and more research could be needed. In general terms, the six-factor model (RMSEA = .04; GFI = .99; AGFI = .99) and a second-factor one (RMSEA = .033; GFI = .99; AGFI = .99) are solutions consistent with previous studies and show that the questionnaire can be considered a reliable (Alphas of the dimensions range from .81 to .94; Omegas from .86 to .97; and mean discrimination of the dimensions from .64 to 88) and useful tool, both in clinical and educational contexts, as well as an instrument for the evaluation of this construct in future research. However, the results of this study also suggest that flow can be explored in greater depth in musicians, taking into account that the writing of the original items was based on the experience of athletes and that the role of time in flow needs to be investigated”.

2.- APA style: better talk about persons, not subjects (see participants section). 

Thank you very much for your advice. The use of persons has been corrected along the text. 

3.- Used only two decimals (see method).

Done 

4.- Add SPSS reference. 

Done 

5.- Please, explain all CFA models tested.

In order to facilitate the reading, we have included a brief description after each model label. For example, if we refer model 1, in brackets we have include “six related factors”. The same for the rest of model labels. The theoretical sense of each model and the references that support them are described in the text but is true that the reading was difficult only with the model labels. Also, we have tried to explain better the models. 

6.- Please test one factor model.

According to the reviewer suggestion we have examined the one factor model. The fit indexes are the following:

Degrees of Freedom = 254

Normal Theory Weighted Least Squares Chi-Square = 6700.39 (P = 0.0)

Satorra-Bentler Scaled Chi-Square = 3722.06 (P = 0.0)

Chi-Square Corrected for Non-Normality = 26692.64 (P = 0.0)

Estimated Non-centrality Parameter (NCP) = 3468.06

90 Percent Confidence Interval for NCP = (3274.39 ; 3669.03)

Minimum Fit Function Value = 10.31

Population Discrepancy Function Value (F0) = 7.97

90 Percent Confidence Interval for F0 = (7.53 ; 8.43)

Root Mean Square Error of Approximation (RMSEA) = 0.18

90 Percent Confidence Interval for RMSEA = (0.17 ; 0.18)

P-Value for Test of Close Fit (RMSEA < 0.05) = 0.00

Expected Cross-Validation Index (ECVI) = 8.77

90 Percent Confidence Interval for ECVI = (8.32 ; 9.23)

ECVI for Saturated Model = 1.38

ECVI for Independence Model = 63.42

Chi-Square for Independence Model with 276 Degrees of Freedom = 27540.03

Independence AIC = 27588.03

Model AIC = 3814.06

Saturated AIC = 600.00

Independence CAIC = 27709.90

Model CAIC = 4047.63

Saturated CAIC = 2123.29

Normed Fit Index (NFI) = 0.86

Non-Normed Fit Index (NNFI) = 0.86

Parsimony Normed Fit Index (PNFI) = 0.80

Comparative Fit Index (CFI) = 0.87

Incremental Fit Index (IFI) = 0.87

Relative Fit Index (RFI) = 0.85

Critical N (CN) = 37.15

Root Mean Square Residual (RMR) = 0.19

Standardized RMR = 0.19

Goodness of Fit Index (GFI) = 0.86

Adjusted Goodness of Fit Index (AGFI) = 0.84

Parsimony Goodness of Fit Index (PGFI) = 0.73

The fit indices are not adequate, may be, because the construct respond to the structure of the theoretical position presented in the paper. Given these results we understand that present this model has not interest and, probably, will make more difficult the reading to potential readers. Nonetheless if the reviewer consider that could be of interest present this model in the paper, please, let us know it.

7.- Please check typo: bifactorial model (bifactor model).

Done

8.- Table 3. Add p values.

According to reviewer suggestion we have included an asterisk in each significant parameter, as is usual reporting results of SEM (a note in each table has been included to explain this issue). We have not included the exact p value because most of them are under .01, then we will need more than two decimals and the tables will contain a lot of information. But the relevant information about the significance of the parameters has been added. 

9.- CFA: add BIC values and IC 90% RMSEA.

According to reviewer suggestion, we have included BIC in the table 2 and the IC for RMSEA along the text where the fit indices of each model are commented. 

10.- Table 4, add p values.

we have included an asterisk in each significant parameter

11.- Please, add CFA measurement invariance analyses.

In all probability the reviewer is considering one of the key elements of an adequate study of validity. In this sense, invariance analysis is necessary to obtain empirical evidences about the construct validity of the scale. If we are considering different populations or groups, and if we suspect that could be found differences in how these groups understand the measured concept, an invariance analysis should be done in order to guarantee that we are measuring with the same conditions (concepts, reliability) between samples. In this research we have not considered conjectures about differences among defined groups as gender, age, instruments, or years of practice, for example. As the reviewer indicate, this could be an interesting issue to be examine, however in this paper, as a first step, our objective was to adapt the FSS-2 to Spanish and to look at the psychometric properties of the instrument in a population composed specifically of musicians. An adequate analysis of the invariance exceed the objective and extension of this paper (another different paper could be done), because an adequate invariance analysis could be extensive in results and discussion if variance is found and firstly we must review the literature to investigate in which variables the multigroup perspective should be considered (gender, instrument, years of practice,…). 

In order to recognize the importance of this issue in the limitations we have added the following text: “…in this sense in order to obtain more validity evidences and be able to generalised the model tested into different groups (gender, instruments used, years of experience,…) an invariance analysis should be explore in forthcoming researches”

Finally, a technical limitation to carry out this kind of analysis is that given the complexity of the tested models the sample size do not is big enough to split it by any moderator variable as gender for example. 

12.- Please, add information about the psychometric properties of: Challenge-skill balance (balance); Clear goals (goals); Unambiguous feedback (feedback). 

According with this suggestion the alphas, omegas and mean discrimination of these dimensions has been added in the instrument section. The new paragraph is:

“…The remaining 3 scales of the FSS-2 that were not included in the EFIM were used as criteria, as they correspond to the dimensions that are necessary conditions in order to generate the flow state. These are: Challenge-skill balance (balance) (in this sample: Alpha = .75; Omega = .83; mean discrimination = .59); Clear goals (goals) (Alpha = .90; Omega = .90; mean discrimination = .78); Unambiguous feedback (feedback) (Alpha = .86; Omega = .87; mean discrimination = .56).”

---

## [Editor Report · Decision Letter 2]

16 Mar 2020

Adaptation to Spanish and psychometric study of the Flow State Scale-2 in the field of musical performers

PONE-D-19-31155R2

Dear Dr. Pablo Holgado-Tello,

We are pleased to inform you that your manuscript has been judged scientifically suitable for publication and will be formally accepted for publication once it complies with all outstanding technical requirements.

With kind regards,

Eduardo Fonseca-Pedrero, PhD

Academic Editor

PLOS ONE